# The delayed clearance of *Talaromyces marneffei* in blood culture may be associated with higher MIC of voriconazole after antifungal therapy among AIDS patients with talaromycosis

**Pengle Guo**, **Wanshan Chen**, **Shaozhen Chen**, **Meijun Chen, Fengyu Hu, Xiejie Chen, Weiping Cai, Xiaoping Tang** *, **Linghua Li** *

Guangzhou Eighth People's Hospital, Guangzhou Medical University, Guangzhou, Guangdong Province, China

☯ These authors contributed equally to this work.
* tangxiaopinggz@163.com (XT); llheliza@126.com (LL)

## Abstract

### Objectives

This study aimed to investigate the influencing factors of delayed clearance of *Talaromyces marneffei* (*T. marneffei*) in blood culture of patients with acquired immune deficiency syndrome (AIDS) complicated with talaromycosis after antifungal therapy.

### Methods

The patients with AIDS complicated with talaromycosis were retrospectively enrolled, and divided into two groups according to the blood *T. marneffei* culture results in two weeks after antifungal therapy. The baseline clinical data were collected and the antifungal susceptibility of *T. marneffei* was tested.

### Results

A total of 190 patients with AIDS and talaromycosis were enrolled, of whom 101 cases remained positive for *T. marneffei* (Pos-group) while the other 89 cases were negative in blood culture (Neg-group) after two weeks' antifungal treatment. The Pos-group had a higher baseline Aspartate aminotransferase (AST, 78.5 vs. 105 U/L; $P = 0.073$) and lower CD4+ T cells level (11 vs. 7 cells/µl; $P = 0.061$). The percentage of isolates with higher MICs of voriconazole (VOR) and fluconazole (FLU) in the Pos-group were significantly higher than those in the Neg-group ($\chi^2 = 12.623$, $P < 0.001$ and $\chi^2 = 9.356$, $P = 0.002$, respectively). By multivariate logistic regression, the MIC value for VOR was identified as the prognostic variable that may influence the clearance of *T. marneffei* in blood culture after antifungal therapy among AIDS patients with talaromycosis.

**Data Availability Statement:** All relevant data are in the manuscript and its Supporting information files.

**Funding:** The study was supported by Guangzhou basic research program on people's Livelihood Science and technology (202002020005), Basic and applied basic research project jointly funded by hospital (College) of Guangzhou (202201020285 and 202201020276) and Medical Key Discipline Program of Guangzhou-Viral Infectious Diseases (2021-2023). The funders had no role in study design, data collection and analysis, decision to publish, or preparation of the manuscript.

**Competing interests:** The authors have declared that no competing interests exist.

## Conclusions

The delayed negative conversion of blood *T. marneffei*-culture may be associated with some factors especially higher MIC of VOR, indicating the possibility of drug resistance of *T. marneffei*.

### Author summary

Slow fungal clearance after antifungal therapy was observed in a few talaromycosis Marneffei (TSM) patients, leading to a significant prolongation of hospital day and increasing adverse drug reaction. The reasons of delayed clearance of Talaromyces Marneffei(*T. marneffei*) is still unclear. Although no standard cutoff value about minimum inhibitory concentration (MIC), the drug sensitivity results are still used to guide the clinical practice. Several studies had reported the MIC value of *T. marneffei* isolates against echinocandin, amphotericin B, and azoles in vitro in the past decade. Patients who infected with azole-resistant strains had a higher mortality risk than those infected with susceptible strains. To response to this phenomenon, the authors hypothesized that the susceptibility of *T. marneffei* strains to antifungal agents could affect its clearance in blood, and verifications were performed in the study.

## 1. Introduction

*Talaromyces marneffei* (*T. marneffei*), previously named *Penicillium marneffei*, is a temperature-dependent dimorphic fungus that causes potentially fatal systemic mycosis (talaromycosis) in immunocompromised patients; and affects people in Southeast Asia and southern China. With the increase of HIV infection rate and immunosuppressant usage, the prevalence of talaromycosis has also greatly increased [1,2]. Talaromycosis is an important opportunistic mycosis in HIV-positive patients, affecting 4–16% of AIDS inpatients in endemic areas [3,4]. In northern Thailand, talaromycosis has become one of the three most common opportunistic diseases, following tuberculosis and cryptococcosis. In southern China, talaromycosis became the main threat among AIDS-related opportunistic disseminated systemic infections [4]. At present, amphotericin B (AMB) followed by itraconazole (ITRA) is the main treatment strategy for talaromycosis [5], and voriconazole (VOR) is regarded as an alternative in patients with AMB intolerance [6,7]. However, the mortality of talaromycosis remains as high as 30% in some regions, representing an important public health problem threatening patients with AIDS [8,9]. The underlying mechanism of poor therapeutic effects in patients with AIDS complicated with talaromycosis is still unclear. In this study, we analyzed the influencing factors of delayed clearance of *T. marneffei* in blood culture after two weeks' antifungal therapy, including the clinical features and the antifungal susceptibilities of *T. marneffei* in vitro, by testing the minimal inhibition concentrations (MICs) of the major antifungal agents [10–12].

## 2. Materials and methods

### 2.1. Ethics statement

This study was approved by the Ethics Review Board of the Guangzhou Eighth People's Hospital (Approval No.202034167). Written informed consent was obtained from all the participants.

## 2.2. Patients

The participants were selected from hospitalized patients in the Infectious Department of Guangzhou Eighth People's Hospital between January 2013 and December 2016. The inclusion criteria were as follows: 1) patients who were diagnosed with HIV infection [13]; 2) at least 18 years old; 3) positive blood culture for *T. marneffei* within three days after admission, and confirming talaromycosis according to Chinese guidelines for diagnosis and treatment of HIV/AIDS [13]; 4) antifungal treatment initiated immediately after talaromycosis diagnosis and lasting for more than two weeks; and 5) blood culture performed every week after antifungal treatment. Conversely, the exclusion criteria included: 1) pregnant or lactating women; and 2) patients with less than 14 days of hospitalization or antifungal treatment.

## 2.3. Data collection

We selected some factors that may affect the clearance of *T. marneffei* strains in vivo including some epidemiologocal data and clinical features, listed as followed.

Epidemiological data, including gender, age, occupation, place of disease onset, and route of HIV infection, were collected.

The following clinical features were also collected: time gap between disease onset to diagnosis, other opportunistic infections, a complete blood count, blood chemistries, CD4+T cell count, CD8+T cell count, the duration of antifungal therapy, and the antifungal regimen.

## 2.4. Study design

This was a retrospective case control study. All enrolled patients were divided into two groups according to the blood *T. marneffei* culture results at two weeks after antifungal therapy [negative group (Neg-group) with negative result and delayed negative group (Pos-group) with positive result]. The baseline clinical data of the two groups were collected and the antifungal susceptibility was tested. Multivariate logistic regression analysis was used to analyze the independent factors that influence the clearance of *T. marneffei* in blood culture after antifungal therapy among AIDS patients with talaromycosis. This study was approved by the Ethics Review Board of the Guangzhou Eighth People's Hospital (Approval No.202034167). Written informed consent was obtained from all the participants.

## 2.5. Culture and identification of *T. marneffei*

Blood samples were collected from patients and placed in BATECFX automatic blood culture instrument (Becton, Dickinson and Company, Franklin Lakes, NJ, USA). The blood samples were inoculated into two Sarpaul solid culture plates (9 cm in diameter, BioMérieux, Marcy-l'Étoile, France) when reported positive, and the two samples were incubated separately in 28˚C and 37˚C incubators. The colony morphology was observed under the microscope. Internal transcribed spacer (ITS) was amplified with polymerase chain reaction (PCR) and sequenced to determine the species when morphological observation alone was insufficient for fungus identification [14].

## 2.6. Antifungal susceptibility testing

Antifungal susceptibility was tested with the Sensititre YeastOne YO10 assay (Thermo Fisher Scientific, Cleveland, OH, USA) according to the manufacturer's instructions. The 96-well Sensititre plates incorporated with alamarBlue for colorimetric determination contained serial two-fold dilutions of the following dried antifungal agents: 0.015–8 μg/mL of anidulafungin (ANI), 0.008–8 μg/mL of micafungin (MICA) and caspofungin (CAS), 0.008–8 μg/mL of

posaconazole (POS) and (VOR), 0.015–16 μg/mL of ITR, 0.12–256 μg/mL of FLU, and 0.12–8 μg/mL of AMB.

Prior to testing, the *T. marneffei* colony was adjusted to 0.5 McFarland yeast suspension, then 20 μL of the yeast suspension was transferred to 11 mL of fungal drug sensitivity inoculation broth, and the inoculation density was $(1.5–8) \times 10^3$ CFU / mL. A total of 100 μL of the yeast suspension was dispensed into each well of the dried panels, and the plates were sealed and placed at 35˚C in a non-$CO_2$ atmosphere. Quality control strains of Candida parapsilosis ATCC 22019 were included throughout the experiments. The MIC value was determined when the color changed from blue (with growth) to red (without growth) after incubation.

### 2.7. Antifungal treatment

All the patients hospitalized received prophylactic treatment with trimethoprim–sulfamethoxazole against *Pneumocystis jiroveci* pneumonia. Patients with talaromycosis received intensive intravenous treatment for 14 days with AMB deoxycholate (purchased from Hebei, China) at a dose of 0.5–0.7 mg/kg per day, or VOR (purchased from Sichuan, China) at a dose of 6 mg/kg twice daily for the first day followed by 4 mg/kg twice daily, or ITRA (purchased from Shanxi, China) at a dose of 200 mg twice daily. Thereafter, all the patients received ITRA capsule at a dose of 200 mg twice daily as maintenance treatment for 10 weeks, followed by prophylactic treatment using ITRA at a dose of 200 mg per day until their CD4$^+$ cell counts reached over 100 cells/μL more than six months after receiving antiretroviral therapy (ART).

### 2.8. Statistical analysis

The data were analyzed using IBM SPSS 22.0 software package. Independent t-test was used to analyze the measurement data in accordance with normal distribution, and rank sum test was used to analyze the data with non-normal distribution. The counting data were analyzed using chi-square test and Fisher's exact test. The distribution of MIC value between the two groups was compared by nonparametric test of ordered variables. Binary logistic regression analysis was used to determine the independent factors that influence the clearance of *T. marneffei* in blood culture after antifungal therapy among AIDS patients with talaromycosis. Cumulative analysis was carried out to assess negative conversion rate of blood cultured fungi The difference was considered to be significant when $P < 0.05$.

## 3. Results

### 3.1. Baseline characteristics of the study population

According to the inclusion and exclusion criteria, a total of 190 patients with AIDS and talaromycosis were enrolled. Among them, 101 cases had positive blood culture results of *T. marneffei* after two weeks of antifungal treatment (Pos-group). The other 89 cases were negative for *T. marneffei* culture after two weeks of antifungal treatment (Neg-group). The subject disposition in the study was shown in Fig 1. Among the 190 patients, 36 (18.9%) patients were co-infected with Pneumocystis pneumonia (PCP), 27 (14.2%) with tuberculosis (TB), 25 (13.2%) with Hepatitis B virus (HBV) infection, and 8 (4.2%) with Hepatitis C virus (HCV) infection. The demographic and baseline clinical characteristics of the 190 patients with AIDS complicated with talaromycosis were listed in Table 1. There were no significant differences regarding age, sex, ART history, and concurrent opportunistic infections (OIs) between the two groups. AMB, ITRA, or VOR were used as antifungal agents, and the therapeutic regimen exhibited no significant difference between the two groups. Common clinical

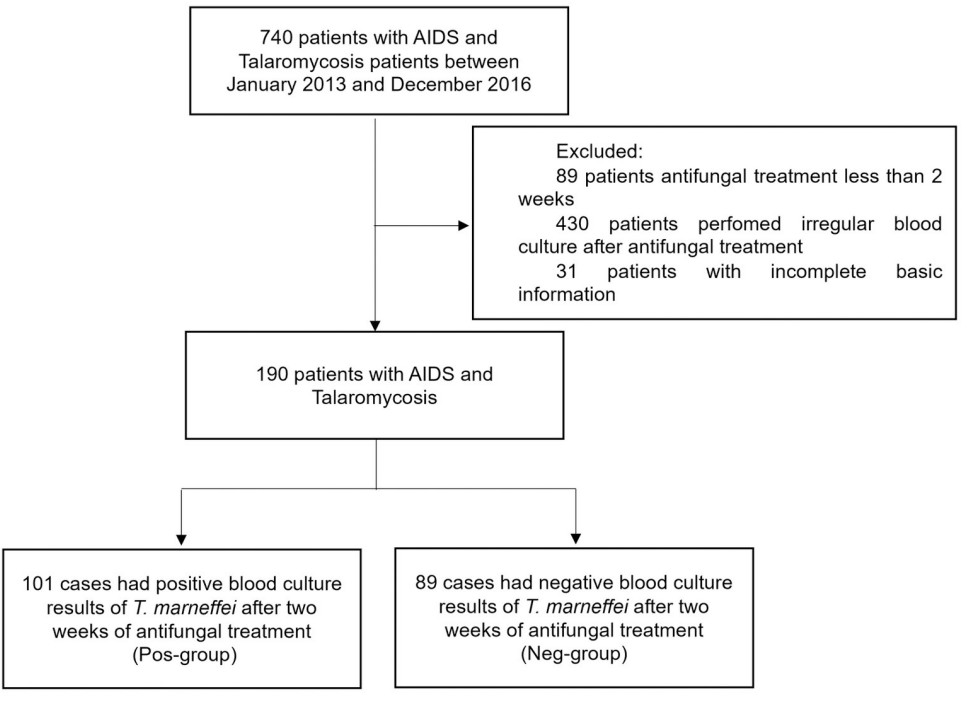

**Fig 1. The subject disposition in the study.**

abnormal signs included peak body temperature, chest imaging abnormality, hepatomegaly, and splenomegaly, and common laboratory abnormal indicators were not significantly different between the two groups. While the Pos-group had a higher baseline Aspartate aminotransferase (AST, 78.5 vs. 105 U/L; $P$ = 0.073) and lower CD4$^+$ T cells level (11 vs. 7 cells/µl; $P$ = 0.061) (shown in Table 1).

## 3.2. Antifungal drug susceptibility of 190 T. marneffei isolates

Strains from the 190 patients were collected and the profiles of antifungal susceptibility at the yeast phase were generated using the Sensititre YeastOne YO10 assay (shown in Fig 2). The baseline MICs from low-to-high were as follows: POS and VOR, ≤0.008−0.06 µg/mL; ITRA, ≤0.015−0.03 µg/mL; AMB, ≤0.25−1 µg/mL; ANI, 4−8 µg/mL; CAS, 2−8 µg/mL; MICA, >8 µg/mL; and FLU, 1−16 µg/mL.

## 3.3. Comparison of antifungal drug susceptibility between two groups

Significant differences were found in the overall distribution of MIC value between the two groups both for VOR ($Z$ = 1.762, $P$ < 0.001) and FLU ($Z$ = 1.211, $P$ = 0.006). The MICs of VOR and FLU in the Pos-group were significantly higher than those in the Neg-group (the proportion of isolates with MICs above 0.008 for VOR: 69/101 (68.3%) vs. 38/89 (42.7%), respectively, $\chi^2$ = 12.623, $P$ < 0.001; the proportion of isolates with MICs above 4 for FLU: 28/101 (27.7%) vs. 9/89 (10.1%), respectively, $\chi^2$ = 9.356, $P$ = 0.002) (shown in Table 2). No differences were found in the distribution of MIC for POS, ITR, AMB, and echinocandins between the two groups.

**Table 1. Demographic and baseline clinical characteristics of 190 patients with AIDS and talaromycosis.**

| | Neg-group | Pos-group | statistical values | P-values |
|---|---|---|---|---|
| Age, years Mean ± SD | 37.3 ± 10.5 | 38.3 ± 13.0 | $t = -0.657$ | 0.512 |
| gender | | | $\chi^2 = 0.069$ | 0.859 |
| Female, n (%) | 18 (20.2%) | 22 (21.8%) | | |
| Male, n (%) | 71 (79.8%) | 79 (78.2%) | | |
| ART history, n (%) | 24 (26.9%) | 26 (25.7%) | $\chi^2 = 0.037$ | 0.848 |
| History with foundational diseases, n (%) | 22 (24.7%) | 19 (18.8%) | $\chi^2 = 0.357$ | 0.602 |
| Concurrent opportunistic infection, n (%) | 30 (33.7%) | 32 (31.7%) | $\chi^2 = 0.088$ | 0.877 |
| Tuberculosis | 14 | 13 | $\chi^2 = 0.317$ | 0.573 |
| Pneumocystis pneumonia | 16 | 20 | $\chi^2 = 0.103$ | 0.749 |
| Coinfection with HBV | 14 | 11 | $\chi^2 = 0.97$ | 0.325 |
| Coinfection with HCV | 3 | 5 | $\chi^2 = 0.293$ | 0.588 |
| Peak body temperature, ˚C Mean ± SD | 39.2 ± 1.0 | 39.3 ± 0.9 | $t = -0.158$ | 0.874 |
| Abnormal Chest imaging, n (%) | 84 (94.4%) | 96 (95.0%) | $\chi^2 = 0.042$ | 0.545 |
| Hepatomegaly, n (%) | 33 (37.1%) | 47 (46.5%) | $\chi^2 = 1.735$ | 0.239 |
| Splenomegaly, n (%) | 47 (52.8%) | 65 (64.4%) | $\chi^2 = 2.607$ | 0.139 |
| Antifungal regimen | | | $\chi^2 = 0.228$ | 0.892 |
| Amphotericin B, n (%) | 47 (52.8%) | 51 (50.5%) | | |
| Voriconazole, n (%) | 30 (33.7%) | 34 (33.7%) | | |
| Itraconazole, n (%) | 12 (13.5%) | 16 (15.8%) | | |
| CD4+count, cells/μl Median (range) | 11 (1–238) | 7 (1–157) | $Z = 3.551$ | 0.061 |
| CD8+ count, cells/μl Median (range) | 220 (4–1080) | 203.5 (11–2609) | $Z = -0.024$ | 0.981 |
| WBC (×10$^9$/L) Mean ± SD | 4.1 ± 2.2 | 4.4 ± 2.6 | $t = 0.425$ | 0.54 |
| Hb (g/L) Mean ± SD | 89.9 ± 22.3 | 88.5 ± 22.2 | $t = 0.143$ | 0.83 |
| PLT (×10$^9$/L) Mean ± SD | 130.3 ± 103.2 | 106.2 ± 78.5 | $t = 1.391$ | 0.15 |
| TBIL μmol/L Median (range) | 9.9 (2.9–105.3) | 10.8 (3.82–186.3) | $Z = -0.872$ | 0.383 |
| ALT U/L Median (range) | 39 (10–485) | 39 (6–586) | $Z = -0.16$ | 0.873 |
| AST U/L Median (range) | 78.5 (17–680) | 105 (11–821) | $Z = -1.796$ | 0.073 |
| LDH U/L Median (range) | 471 (47–6060) | 513 (170–5831) | $Z = -0.796$ | 0.426 |
| SCr μmol/L Median (range) | 70.1(35–242.6) | 66 (30.3–318) | $Z = -1.355$ | 0.175 |
| PCT ng/ml Median (range) | 0.7 (0.1–25.8) | 1.2 (0.1–44.3) | $Z = -1.451$ | 0.147 |
| Lac mmol/l Median (range) | 1.4 (0.5–10) | 1.7 (0.5–8.2) | $Z = -0.865$ | 0.387 |

Abbreviations: SD, standard deviation; ART, antiretroviral therapy; WBC, white blood cell; ALT, alanine aminotransferase; AST, aspartate aminotransferase; TBIL, total bilirubin; Hb, hemoglobin; PLT, platelet; LDH, lactate dehydrogenase; SCr, Serum creatinine clearance rate; PCT, Procalcitonin; Lac, lactic acid.

### 3.4. Multivariate analysis for the influencing factors of the clearance of *T. marneffei* in blood culture after antifungal therapy among AIDS patients with talaromycosis

Multivariate analysis using the logistic regression model involving the 2 significant factors (AST and CD4+T levels)determined by univariate analysis and the MIC values for VOR and FLU. The variable assignment was shown in Table 3. The MIC value for VOR was identified as the prognostic variable derived from binary logistic regression analysis that influence the clearance of *T. marneffei* in blood culture after antifungal therapy among AIDS patients with talaromycosis (shown in Table 4).

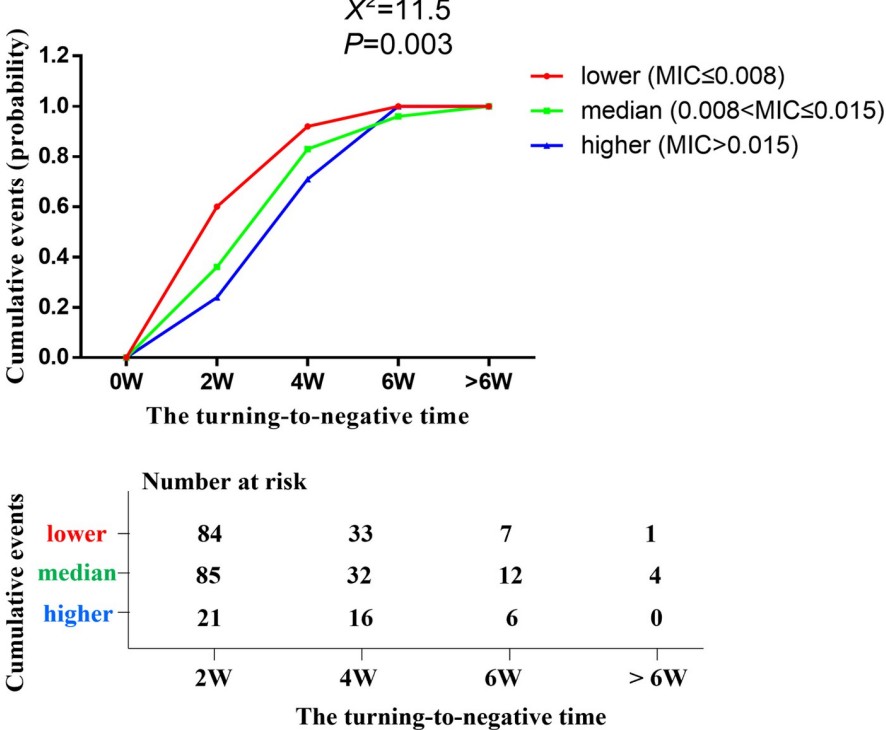

$X^2 = 11.5$
$P = 0.003$

**Fig 2. The MIC values of voriconazole might affect the clearance of *T. marneffei* in blood cultures among AIDS patients with talaromycosis after antifungal therapy.** Strains from 190 patients were split into three groups according to the MIC values for voriconazole *in vitro*: lower (MIC ≤ 0.008), medium (0.008 < VOA ≤ 0.015), and higher (VOA > 0.015). Cumulative negative conversion rate of blood cultured fungi was assessed among three groups after 2, 4, 6 weeks and more than 6 weeks of antifungal treatment. $P < 0.05$ is considered statistically significant.

**Table 2. Antifungal drug susceptibility of *T. marneffei* strains from 190 patients with AIDS and talaromycosis.**

| Antifungal agents | group | MIC (µg/ml) No. of patients(%) | | | | | | | | | | Z | P |
|---|---|---|---|---|---|---|---|---|---|---|---|---|---|
| | | 0.008 | 0.015 | 0.03 | 0.06 | 0.25 | 0.5 | 1 | 2 | 4 | > 8 | | |
| posaconazole | Neg-group | 81 (91%) | 7 (7.9%) | 0 | 1 (1.1%) | | 0 | 0 | | 0 | 0 | 0.07 | 1 |
| | Pos-group | 94 (93.1%) | 7 (6.9%) | 0 | 0 | | 0 | 0 | | 0 | 0 | | |
| voriconazole | Neg-group | 51 (57.3%) | 33 (37.1%) | 3 (3.4%) | 2 (2.2%) | | 0 | 0 | | 0 | 0 | 1.762 | **<0.001** |
| | Pos-group | 32 (31.7%) | 53 (52.5%) | 12 (11.9%) | 4 (3.9%) | | 0 | 0 | | 0 | 0 | | |
| itraconazole | Neg-group | 0 | 87 (97.8%) | 2 (2.2%) | 0 | | 0 | 0 | | 0 | 0 | 0.118 | 0.686 |
| | Pos-group | 1 (1.0%) | 96 (95.0%) | 4 (4.0%) | 0 | | 0 | 0 | | 0 | 0 | | |
| Amphotericin B | Neg-group | | | | | 11 (12.4%) | 59 (66.3%) | 19 (21.3%) | | 0 | 0 | 0.58 | 0.291 |
| | Pos-group | | | | | 21 (20.8%) | 61 (60.4%) | 19 (18.8%) | | 0 | 0 | | |
| fluconazole | Neg-group | | | | | | | 2 (2.2%) | 22 (24.8%) | 56 (62.3%) | 9 (10.1%) | 1.211 | **0.006** |
| | Pos-group | | | | | | | 3 (3.0%) | 16 (15.8%) | 54 (53.5%) | 28 (27.7%) | | |

**Table 3. Variable assignment included in regression analysis.**

| Factors | Name | assignment |
|---|---|---|
| blood culture results of *T. marneffei* after two weeks of antifungal treatment | Y | Positive = 1, Negative = 0 |
| AST | X1 | continuous variable |
| CD4+T cells | X2 | continuous variable |
| MIC for VOR | X3 | <0.015 = 1, 0.015 = 2, >0.015 = 3 |
| MIC for FLU | X4 | <4 = 1, 4 = 2, >4 = 3 |

**Table 4. Binary logistic regression analysis for the influencing factors of the clearance of Talaromyces marneffei in blood culture after antifungal therapy among AIDS patients with talaromycosis.**

| Variables | b | SE | wald $\chi^2$ value | Sig(P) | OR | 95%CI for OR |
|---|---|---|---|---|---|---|
| CD4 | -0.002 | 0.003 | 0.332 | 0.564 | 0.998 | 0.992~1.004 |
| AST | 0.002 | 0.001 | 2.063 | 0.151 | 1.002 | 0.999~1.004 |
| FLU 1 (MIC<4) | | | 3.097 | 0.213 | | |
| FLU 2 (MIC = 4) | -0.52 | 0.463 | 1.261 | 0.261 | 0.594 | 0.24~1.474 |
| FLU 3 (MIC>4) | 0.337 | 0.784 | 0.185 | 0.667 | 1.401 | 0.302~6.51 |
| VOR 1 (MIC = 0.008) | | | 6.369 | 0.041 | | |
| VOR 2 (MIC = 0.015) | 1.063 | 0.424 | 6.291 | 0.012 | 2.895 | 1.262~6.644 |
| VOR 3 (MIC>0.015) | 1.286 | 0.91 | 1.995 | 0.158 | 3.618 | 0.607~21.547 |
| constant | -0.414 | 0.383 | 1.172 | 0.279 | 0.661 | |

Abbreviations: AST, aspartate aminotransferase; FLU, fluconazol; VOR, Voriconazole. MIC, Minimum inhibitory concentration; B, beta; SE, standard error; OR, odds ratio.

### 3.5. The MIC value of voriconazole may affect the clearance of *T. marneffei* in blood cultures among AIDS patients with talaromycosis after antifungal therapy

Strains from the 190 patients were split into three groups according to their MIC values for voriconazole in vitro: lower (MIC≤0.008), medium (0.008<VOA≤0.015), and higher (VOA>0.015). And cumulative analysis to assess negative conversion rate of blood cultured fungi was carried out among the three groups after 2, 4, 6 weeks and more than 6 weeks of antifungal treatment. Patients in the lower group, medium group, and higher group were 84, 85, and 21, respectively. The cumulative negative conversion rate of blood cultured fungi among three groups after 2, 4, 6 weeks and more than 6 weeks of antifungal treatment was significantly different (Shown in Fig 2), indicating that the MIC values of voriconazole significantly affects the clearance of *T. marneffei* in blood. Similarly, we found the same performance in fluconazole. (Shown in S1 Fig).

## 4. Discussion

Talaromycosis is a systemic disseminated disease with a high mortality rate, partly due to the poor efficacy of antifungal treatments. A randomized clinical trial showed that 32.2% of patients with HIV and talaromycosis still had positive blood cultures for *T. marneffei* by day 8 after initiation of itraconazole treatment [5]. In our study, 101/190 (53.2%) patients had delayed conversion of *T. marneffei* in blood culture after two weeks of antifungal therapy. In

order to understand the influencing factors of delayed clearance of *T. marneffei*, we compared some promising indicators between two groups with clearance or delayed clearance of *T. marneffei* in blood culture after two weeks' antifungal therapy, including the clinical features and the antifungal susceptibilities of *T. marneffei in vitro*. We found that the delayed clearance group had higher AST level and lower CD4+T cells, also with higher MIC values for fluconazole and voriconazole.

It was consistent with the previous study that the elevated serum AST was determined as the independent risk factors for poor outcome in AIDS patients complicated with *T. marneffei* and the poor outcome group had significantly lower CD4+ T cell count [15,16]. In fact, AST, not only a marker of hepatocyte injury, also origins from cells of other tissues, such as brain, myocortical cells and skeletal muscle cells. So the association of AST levels with severe illness or mortality probably involves the development of multiorgan impairment, which is common in severe and death cases of COVID-19[17,18]. Further research to understand the dynamics of AST and CD4+ T cell count and determine whether it is a useful biomarker of talaromycosis.

The MIC for voriconazole was determined as the independent factor by binary logistic regression analysis, which indicates that the susceptibility of *T. marneffei* strain to antifungal agents could affect the antifungal efficacy. The survival analysis of cumulative negative conversion rate of blood cultured fungi further showing that voriconazole sensitivity *in vitro* may affect T. marneffei clearance in the bloodstream. Our study is consistent with some case reports, in which patients who were infected with azole-resistant strains had a higher mortality incidence than patients who are infected with susceptible strains [19,20]. Therefore, when there is a poor response to treatment for talaromycosis in patients with AIDS, attention should be paid to the *T. marneffei* resistance to fluconazole or voriconazole. With the increasing trend of other drug-resistant fungi, *T. marneffei* has gradually developed resistance to fluconazole [10–12]. Thus the sensitivity of *T. marneffei* to voriconazole should be closely monitored.

Our study showed that posaconazole, itraconazole, and voriconazole had good application prospects for the treatment of *T. marneffei* infection due to their low MIC values. Posaconazole has not been yet recommended for the treatment of talaromycosis, but a recent study from the University of Hong Kong showed that it has beneficial effects against *T. marneffei* infection [21].

The susceptibility of *T. marneffei* strain to echinocandins is different at hyphal forms or yeast forms [21,22]. Anidulafungin had some antifungal activity against the hyphal phase of *T. marneffei* strain, with a MIC value of less than 2 μg/mL, but the yeast phase of *T. marneffei* was not sensitive to it [23]. Lau detected the susceptibility of 57 *T. marneffei* strains using the broth microdilution method and the E test method, and found that the MIC values of anidulafungin were beyond the sensitive range [19]. In our study, we also found that the *T. marneffei* strains were not sensitive to the three kinds of echinocandins. Although echinocandins have been recommended for the treatment of candidiasis, it was inferred from this *in vitro* study that these drugs might not be suitable for the treatment of talaromycosis.

As the number of countries reporting resistance against azoles continues to increase, it is more and more difficult to achieve effective antifungal treatment. In order to guide the clinical antifungal therapy, further effort is needed to widely monitor the susceptibility of pathogenic filamentous fungi. Our study found that the decrease of antifungal drug sensitivity of *T. marneffei in vitro* could affect the efficacy of the antifungal treatment, but the mechanism of drug resistance of the *T. marneffei* was unclear. In the future, we will conduct the genome sequencing for the *T. marneffei* strains with poor antifungal susceptibility in order to find the genes related to drug resistance.

Our study has limitations. Our observation time window is relatively short, and it is not clear whether the decline of antifungal drug sensitivity of *T. marneffei* affects the long-term outcome of patients. In addition, As there is no break point to determine the susceptibility and drug resistance of *T. marneffei in vitro*, the relationship between MICs and the antifungal susceptibility of filamentous fungi is needed to be further clarified.

## 5. Conclusion

The delayed negative conversion of blood *T. marneffei*-culture may be associated with some factors especially higher MIC of VOR. Further research are needed to understand the predictive value of AST and CD4+ T cell count in the efficacy evaluation of talaromycosis.

## Supporting information

**S1 Fig. The MIC values of fluconazole might affect the clearance of *T. marneffei* in blood cultures among AIDS patients with talaromycosis after antifungal therapy.** Strains from 190 patients were plit into two groups according to the MIC values for fluconazole in vitro: lower (MIC≤4) and higher (MIC>4). Cumulative negative conversion rate of blood cultured fungi was assessed between the two groups after 2, 4, 6 weeks and more than 6 weeks of antifungal treatment. $P<0.05$ is considered statistically significant.
(TIF)

**S1 Table. The original Data (patient names displayed in digital code) of the manuscript.**
(XLSX)

## Acknowledgments

We would like to thank the patients who participated in our study.

## Author Contributions

**Conceptualization:** Weiping Cai, Xiaoping Tang, Linghua Li.

**Data curation:** Pengle Guo, Wanshan Chen, Shaozhen Chen, Meijun Chen, Fengyu Hu, Xiejie Chen.

**Methodology:** Pengle Guo, Wanshan Chen, Shaozhen Chen, Meijun Chen, Fengyu Hu, Xiejie Chen, Weiping Cai, Xiaoping Tang, Linghua Li.

**Project administration:** Weiping Cai, Xiaoping Tang, Linghua Li.

**Supervision:** Weiping Cai, Xiaoping Tang, Linghua Li.

**Writing – original draft:** Pengle Guo, Xiaoping Tang, Linghua Li.

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
