## [Decision Letter · Decision Letter 0]

12 Dec 2022

Dear PhD. Li,

Thank you very much for submitting your manuscript "The delayed clearance of Talaromyces marneffei in blood culture may be associated with higher MIC of voriconazole after antifungal therapy among AIDS patients with talaromycosis" for consideration at PLOS Neglected Tropical Diseases. As with all papers reviewed by the journal, your manuscript was reviewed by members of the editorial board and by several independent reviewers. In light of the reviews (below this email), we would like to invite the resubmission of a significantly-revised version that takes into account the reviewers' comments. 

We cannot make any decision about publication until we have seen the revised manuscript and your response to the reviewers' comments. Your revised manuscript is also likely to be sent to reviewers for further evaluation.

Sincerely,

Marcio L Rodrigues

Section Editor

Marcio Rodrigues

Section Editor

Reviewer's Responses to Questions

**Key Review Criteria Required for Acceptance?**

**Methods**

-Are the objectives of the study clearly articulated with a clear testable hypothesis stated?

-Is the study design appropriate to address the stated objectives?

-Is the population clearly described and appropriate for the hypothesis being tested?

-Is the sample size sufficient to ensure adequate power to address the hypothesis being tested?

-Were correct statistical analysis used to support conclusions?

-Are there concerns about ethical or regulatory requirements being met?

Reviewer #1: In general the study meets these criteria. Key question was clearly stated and appropriate analysis conducted.

Reviewer #2: All criteria are fulfilled

Reviewer #3: (No Response)

**Results**

-Does the analysis presented match the analysis plan?

-Are the results clearly and completely presented?

-Are the figures (Tables, Images) of sufficient quality for clarity?

Reviewer #1: Some results lack clear description, presentation and interpretation.

Reviewer #2: Results are displayed clearly and completely

Comments 

Figure 1: correct the black background

Figure 2: increase the quality or make a diagram. If possible discard the figure as they do not provide much relevant content.

Reviewer #3: (No Response)

**Conclusions**

-Are the conclusions supported by the data presented?

-Are the limitations of analysis clearly described?

-Do the authors discuss how these data can be helpful to advance our understanding of the topic under study?

-Is public health relevance addressed?

Reviewer #1: Final conclusion was well-stated based on the data.

Reviewer #2: All criteria are fulfilled

Reviewer #3: (No Response)

**Editorial and Data Presentation Modifications?**

Reviewer #1: Please check grammatical errors and revise figures and tables.

Reviewer #2: Excellent methodological and statistical work. Only the figures require minor corrections.

Nothing more to comment

Good work, it was a pleasure to read it.

Reviewer #3: (No Response)

**Summary and General Comments**

Reviewer #1: Generally, the study clearly described the efforts to identify factors impacting fungal infection clearance in HIV positive patients. However, some methods and results need to be better presented and explained.

Reviewer #2: Excellent methodological and statistical work. Only the figures require minor corrections.

nothing more to comment

Good work, it was a pleasure to read it.

Reviewer #3: In this study, Guo et al., investigated the reason behind delayed clearance of Talaromyces marneffei in HIV/AIDS patients. The authors assessed the invitro antifungal activities fungicidal agents against the clinical isolates and efforts were carried out to draw plausible correlations between the observed variations in the delayed clearance of infections. Although this is an interesting study, there are several areas that need to addressed before considering the manuscript for publication. 

1) The MIC values of voriconazole given on the Table 1 for the both positive and Negative groups are within the reported MIC90 values of voriconazole, therefore, it is presumable that the isolates are susceptible to voriconazole and the observed difference is a mere variation among the population and cannot be attributed to resistant phenotype. The authors should consider redoing the statistical analysis with the percentage of strains that has MICs above the MIC90 value. The authors are recommended to do a similar analysis on fluconazole using MIC90. 

2) Table 1 and Page 6(lines 139-140). The authors states that there were no significant differences were evident between the groups with respect to co-infections. It appears that authors looked for plausible correlation between a co-infection and delayed clearance of T. marneffei. Did the authors do the analysis with individual co-infection agent? The authors are suggested to do the analysis with individual co-infection agent to see any potential correlation exists. 

3) The authors have used AST as marker, however increased AST could be due to several reasons. The authors have put little effort to explain the observed co-relation. Some of the co-infection agents could enhance the AST level. Liver is one of the major organs where T. marneffei infects, it is plausible that the extensive liver damage caused by a co-infective agent such as hepatitis B, delayed the clearance of fungi. Also, any kind of chemotherapeutic agents that the patient received could also influence the level of AST.

PLOS authors have the option to publish the peer review history of their article (what does this mean?). If published, this will include your full peer review and any attached files.

Reviewer #1: Yes: Mengyao Niu

Reviewer #2: Yes: Fausto Cabezas-Mera

Reviewer #3: No
---

## [Editor Report · Decision Letter 1]

27 Feb 2023

Dear Dr. Li,

We are pleased to inform you that your manuscript 'The delayed clearance of Talaromyces marneffei in blood culture may be associated with higher MIC of voriconazole after antifungal therapy among AIDS patients with talaromycosis' has been provisionally accepted for publication in PLOS Neglected Tropical Diseases.

Best regards,

Angel Gonzalez, Ph.D.

Academic Editor

Marcio Rodrigues

Section Editor

---

## [Editor Report · Acceptance letter]

29 Mar 2023

Dear PhD. Li,

We are delighted to inform you that your manuscript, "The delayed clearance of Talaromyces marneffei in blood culture may be associated with higher MIC of voriconazole after antifungal therapy among AIDS patients with talaromycosis," has been formally accepted for publication in PLOS Neglected Tropical Diseases.

Best regards,

Shaden Kamhawi

co-Editor-in-Chief

Paul Brindley

co-Editor-in-Chief
